# Associations of Maternal Consumption of Dairy Products during Pregnancy with Perinatal Fatty Acids Profile in the EDEN Cohort Study

**DOI:** 10.3390/nu14081636

**Published:** 2022-04-14

**Authors:** Wen Lun Yuan, Jonathan Y. Bernard, Martine Armand, Catherine Sarté, Marie Aline Charles, Barbara Heude

**Affiliations:** 1Université Paris Cité, Inserm, INRAE, CRESS, F-75004 Paris, France; jonathan.bernard@inserm.fr (J.Y.B.); marie-aline.charles@inserm.fr (M.A.C.); barbara.heude@inserm.fr (B.H.); 2Singapore Institute for Clinical Sciences, Agency for Science, Technology, and Research (A*STAR), Singapore 117609, Singapore; 3Aix Marseille Univ, CNRS, CRMBM, Marseille, France; martine.armand@inserm.fr (M.A.); catherine.sarte@inserm.fr (C.S.)

**Keywords:** dairy product, pregnancy, fatty acids, maternal red blood cells, cord red blood cells, colostrum

## Abstract

Maternal diet is the main source of fatty acids for developing offspring in-utero and in breastfed infants. Dairy products (DP) are important sources of fat in the European population diet. C15:0 and C17:0 fatty acids have been suggested as biomarkers of dairy fat consumption. This study’s aim is to describe the associations between maternal DP (milk included) consumption during pregnancy and C15:0, C17:0 and polyunsaturated fatty acid (PUFA) levels in perinatal biofluids. Study populations were composed of 1763, 1337 and 879 French mothers from the EDEN (“Étude des Déterminants pre- et post-natals de la santé de l’ENfant”) study, with data on maternal and cord red blood cells’ (RBC) membrane and colostrum, respectively. Associations were assessed using linear regression models adjusted for recruitment center, maternal age, healthy dietary pattern or fish consumption. Greater adherence to a ”cheese” consumption pattern was associated with lower linoleic acid level in colostrum and higher C15:0 and C17:0 levels but in a less consistent manner for C17:0 across biofluids. Greater adherence to “semi-skimmed milk, yogurt” and “reduced-fat DP” patterns was related to higher docosahexaenoic acid and total n-3 PUFA levels and lower n-6/n-3 long-chain PUFA ratio in maternal and cord RBC. Our results suggest that C15:0 could be a good biomarker of maternal dairy fat consumption in perinatal biofluids.

## 1. Introduction

Long-chain polyunsaturated fatty acids (LC-PUFAs), in particular eicosapentaenoic acid (EPA, C20:5 n-3), docosahexaenoic acid (DHA, C22:6 n-3) and arachidonic acid (AA, C20:4 n-6), contribute to placental function and to the development of the offspring’s brain and retina [1,2]. Fetus and breastfed infant exposure to LC-PUFAs depends on maternal dietary LC-PUFAs intake, adipose tissue storage and metabolism via the PUFAs biosynthesis pathway [3]. Indeed, LC-PUFAs are also synthetized, mainly by the liver, through series of desaturation and elongation from precursors, linoleic (LA, C18:2 n-6) and alpha- linolenic (ALA, C18:3 n-3) acids, named essential fatty acids because they can only be provided by diet [3,4]. LC-PUFAs are preferentially transported from maternal circulation to the fetus through the placenta. After delivery, maternal LC-PUFAs are also expressed in breastmilk and transferred to the breastfed offspring [3,5,6,7].

Excessive amount of dietary LA over a low dietary ALA intake reduces the conversion of ALA to EPA and DHA due to a competition between n-6 and n-3 PUFAs for the same elongases and desaturases [8,9]. LA and ALA are mainly provided by vegetable oils, nuts and seeds consumption. Evidence has suggested that greater maternal LA intake during pregnancy was associated with higher LA level in colostrum but not necessarily in cord blood [10,11]. In France, dairy products (DP) are the second largest contributor to adult fat intake [12]. Interestingly, dairy fat is the most complex natural fat as it contains approximately 400 different fatty acids [13]. Evidence from an animal study has shown that having a diet with dairy lipids lead to higher n-3 LC-PUFAs levels in several tissues. Hence, it was suggested that short- and medium-chain fatty acids from dairy lipids consumption may be preferentially β-oxidized compared with ALA, which may favor the conversion pathway of ALA to n-3 LC-PUFAs [14]. Randomized controlled trials have suggested that greater maternal consumption of dairy fat was associated with lower LA level in breast milk [15,16]. So far, only one observational study conducted on 55 Swedish mothers found supporting results, but dairy products consumption was evaluated using 24-h recalls and 24-h food diaries [17].

Pentadecanoic acid (C15:0) and heptadecanoic acid (C17:0) levels, mainly in plasma phospholipids and red blood cells (RBC) membrane, have been used in numerous studies as biomarkers of dairy fat consumption [18,19,20]. Indeed, these odd-chain saturated fatty acids (OCFAs) originate from rumen microbial fermentation and are excreted in the milk [13]. Another dietary source of OCFAs is fish and some evidence suggested that C17:0 can be marginally derived from fiber-rich foods [21,22,23,24,25]. Results from animal studies have suggested that dietary OCFAs are capable of passing through the placental barrier and into the milk of mammals [26,27]. Yet, no study has been conducted in humans to confirm this finding. In the present study, using data from the EDEN mother–child cohort study, we aimed to describe maternal DP consumption during pregnancy and their associations with OCFAs and PUFAs levels in maternal and cord RBC membrane and colostrum. Here, we hypothesized that C15:0 and C17:0 in maternal or cord blood and colostrum are biomarkers of maternal dairy fat consumption during pregnancy hence that C15:0 and C17:0 could be transferred to the fetus and the breastfed infants. Second, we hypothesized that maternal high-fat content dairy products consumption could influence n-6 and n-3 precursors and LC-PUFAs levels (through the modulation of the precursors conversion rate to LC-PUFAs) and hence, the n-6/n-3 LC-PUFA ratio in all perinatal biofluids.

## 2. Materials and Methods

### 2.1. Study Population

This study was conducted within the EDEN (“Étude des Déterminants pre- et post-natals de la santé de l’ENfant”) study, an ongoing bicentric French mother-child cohort. The detailed study design has been previously described [28]. Briefly, in two maternity hospital units, in Nancy and Poitiers, 2002 pregnant women were recruited during their hospital visit before 24 weeks of gestation, between 2003 and 2006. Women were eligible if they had a singleton pregnancy, no diabetes prior to pregnancy, no intention of moving out of the city within the following 3 years and were capable of reading and writing in French. For the present analyses, study populations were restricted to women with complete data for DP consumption during pregnancy, who provided perinatal biological samples afterwards and whom biological samples had a valid fatty acid composition, i.e., all fatty acid level < 4 SD in the biofluids (*n* = 1754 with maternal blood sample, *n* = 1337 with cord blood sample, *n* = 879 with colostrum sample), as detailed in Figure 1.

### 2.2. Maternal DP Consumption

At birth, maternal diet over the last 3 months of pregnancy was assessed using a self-administered validated food frequency questionnaire (FFQ) [29]. This FFQ was composed of 137 items with 7 categories of frequencies ranging from “never” to “more than once a day”. Food frequency were subsequently converted to be expressed on a daily scale. Hence, DP comprised milk (3 items), cheese (7 items), yogurt (3 items) and added dairy fats (4 items). Using frequencies of DP consumption and principal component analysis, three DP consumption patterns (“cheese”, “reduced-fat DP”, “semi-skimmed milk, yogurt”) were derived, explaining 33% of the total variance (as described in Table 1). The number of patterns retained were selected based on the scree plot and the interpretability of the patterns. These patterns provide a holistic picture of DP consumption. However, since DP patterns may differ from one population to another, this data-driven approach limits future in between studies comparison. We hence had a secondary analysis using a quantitative evaluation of DP consumption by calculating maternal daily DP consumption in grams. In the FFQ, usual serving sizes were reported for milk consumption using pictures extracted from SUVIMAX (“SUpplémentation en VItamines et Minéraux Anti-oXydants”) validated pictures booklet [30]; otherwise, for remaining DP, middle size servings were assigned based on SUVIMAX booklet. For each food item, serving sizes were multiplied by the consumption frequency to obtain grams per day. As we were interested in distinguishing dairy fat content within DP, we summed and grouped cheese and added dairy fats into “higher-fat content” DP and milk and yogurt into “lower-fat content” DP. Due to the skewed distribution of daily consumption of higher-fat and lower-fat content DP, in favor of heavy consumers, consumers were grouped into tertiles (“low consumer”, “moderate consumer”, “heavy consumer”). As there were less than 2% of non-consumer of higher-fat or lower-fat content DP, non-consumers were grouped into “low consumer”.

### 2.3. Biofluids Fatty Acids Composition Assessment

Maternal fasting and cord blood samples were collected at 24–28 weeks of gestation and at delivery, respectively. RBCs membrane were isolated by centrifugation [31] and stored at −80 °C until analysis. Colostrum samples of about 5 mL were collected at the hospitals during the first week after delivery from one feed by manual expression and were stored at −80 °C until analysis. RBC membrane and the colostrum fatty acids composition was assessed by gas chromatography following procedures published elsewhere [10]. Briefly, a direct methylation procedure was performed on 50 µL of RBC membranes or 100 µL of colostrum samples at 100 °C for 1 h using methanol/hexane and acetyl chloride. Fatty acid methyl-esters (FAME) were analyzed by gas chromatography (Clarus 680, PerkinElmer, Waltham, MA, USA), flame ionization detector, Totalchrom software 6.3 (PerkinElmer, Waltham, MA, USA), hydrogen as gas carrier) using a fused silica capillary fast column (BPX 70, 10 m × 0.1 mm i.d., 0.2 mm film thickness (Sigma-Supelco, Bellefonte, PA, USA)). Each fatty acid level was expressed as the proportion of total fatty acids present in the chromatogram (weight percent).

### 2.4. Covariates

During the 24–28 weeks of gestation, a face-to-face interview, maternal age and pre-pregnancy weight were reported and maternal height was measured. Information about maternal gestational diabetes was obtained from obstetrical records. Gestational age at delivery was determined from the date of the last menstrual period and early standard ultrasound fetal measurement. Maternal “healthy” dietary pattern during pregnancy was derived previously from the aforementioned FFQ [32]. The frequency of fish consumption during pregnancy was extracted from the FFQ based on 5 items (fresh fish, oil-preserved fish, smoked or salt-preserved fish, breaded fish, fish-based dish).

### 2.5. Statistical Analysis

We described all variables using univariable statistics, i.e., means/standard deviations and percentages. Characteristics of each study population that potentially undergo a differential participants retention was tabulated. Comparison between included and excluded participants was performed using Student *t*-test and Chi-square tests in Appendix A.

Analyses on the association between maternal DP consumption and perinatal biofluids fatty acids levels were run using non-adjusted and adjusted linear regression. Fatty acids level was standardized to enable comparison between their regression coefficients. Three models were successively built using either DP consumption patterns or DP consumption tertiles as exposure. When analyzing DP consumption patterns, all patterns were included simultaneously in each model, since they are independent from each other by design. Similarly, “higher-fat content” and “lower-fat content” DP consumption tertiles were studied simultaneously within the same model. Model 1 included adjustments for non-dietary covariates (study center, sampling day (for colostrum analysis), maternal age at delivery). “Higher-fat content” DP (except butter) consumption contributes greatly to maternal “healthy” dietary pattern, similar to fish and fruits and vegetables consumption, that are also potential sources of OCFAs [32]. Hence, greater “higher-fat content” DP consumption associations could be partly explained by having a healthier diet in overall. For these reasons, model 2 was additionally adjusted for maternal “healthy” dietary pattern during pregnancy. To distinguish the important contribution of fish consumption to LC-PUFAs and possibly to OCFAs intake but not as part of a healthy diet, model 3 was adjusted on model 1 covariates and for the frequency of fish consumption during pregnancy (instead of maternal “healthy” dietary pattern as in model 2).

Interaction of maternal preconception overweight status on our associations was tested as fatty acid metabolism might be altered in overweight (and obese) individuals [33]. Gestational diabetes and hypertensive disorders is also known to impair fatty acid metabolism [34]. As only 6 and 5% of mothers in our study samples had gestational diabetes or hypertensive disorders, respectively, we removed these mothers in our sensitivity analyses. We also excluded mothers with extreme values for energy intake (<1000 kcal/day or >5000 kcal/day) reflecting under- or over-reporters that could yield or mask spurious associations. Finally, we further removed preterm delivery (defined as a gestational age < 37 weeks), in our analyses regarding cord RBC and colostrum, as it could affect fatty acids levels in the aforementioned perinatal biofluids [35,36]. Significance level was set at alpha = 0.05 for all tests except for interaction test (alpha = 0.10). No imputation on the outcomes, the exposures, the confounders and the moderator (missing data = 1–2% for covariates) was performed. As all our tested associations were hypothesis driven and as the studied fatty acids are not independent of each other (% of total fat level in a given biofluid), we did not perform any correction for multiple testing.

Analyses were performed using SAS (version 9.4; SAS Institute, Cary, NC, USA) and forests plots were obtained using R software (version 4.1.1; R Core Team (2017). R: A language and environment for statistical computing. R Foundation for Statistical Computing, Vienna, Austria. URL https://www.R-project.org/, accessed on 16 March 2022). The analysis plan was pre-registered online on OSF (Open Science Framework) [37].

## 3. Results

Study populations characteristics are presented in Table 2. On average, in all study populations, mothers were aged 29 (± 5) years old. About one third of the mothers had a university degree higher than 2 years. One quarter of the mothers were overweight before pregnancy (obesity included), smoked during pregnancy or breastfed at least 6 months. There was no substantial difference between the characteristics of the participants with maternal blood and those with cord blood. Study populations on maternal or cord RBC membrane composition did not differ largely with the EDEN full cohort, even if included participants had higher educational attainment (Appendix A). Compared with the other study populations and EDEN full cohort, participants that provided a colostrum sample were more likely to be highly educated, have higher household income, have lower body mass index before pregnancy, were a nonsmoker during pregnancy and have longer breastfeeding duration. Same differences were observed once we compared characteristics of included and non-included participants (Appendix A). Importantly, no large difference in DP consumption was observed between study populations (Appendix A).

In overall, for the same fatty acids, levels across perinatal biofluids were positively correlated (Appendix A). Maternal ALA level was therefore negatively correlated with the one in cord RBC membrane. The levels of C15:0 and C17:0 in cord RBC membrane did not correlate with the ones in colostrum. Maternal RBC membrane, cord RBC membrane and colostrum OCFAs and PUFAs levels are presented in Appendix A. C15:0 and ALA levels were higher in colostrum than in maternal or cord RBC membrane. C17:0 level was similar across the three perinatal biofluids. DHA and total n-3 PUFA levels were lower in colostrum than in RBC membrane. Similarly, AA level was lower in colostrum than in maternal or cord RBC membrane. LA level in cord RBC membrane was the lowest. Total n-6 PUFA level and the n-6/n-3 LC-PUFA ratio were lower in colostrum than in RBC membrane.

### 3.1. Associations between Maternal DP Consumption and Fatty Acids Levels in Maternal RBC Membrane

Associations between maternal DP patterns and fatty acids levels in maternal RBC membrane are shown in Figure 2. Greater adherence to “cheese” consumption pattern was associated with higher C15:0, C17:0, and ALA level across all models. Greater adherence to “cheese” consumption pattern was also related to higher DHA and total n-3 PUFA levels and to lower total n-6 PUFA and n-6/n-3 LC-PUFA ratio; however, they did not remain significant after adjustment for either maternal healthy dietary patterns or fish consumption during pregnancy. Higher adherence to “reduced-fat DP” consumption pattern was associated with lower C15:0, C17:0, LA, total n-6 PUFA levels and n-6/n-3 LC-PUFA ratio; and with higher ALA, DHA and total n-3 PUFA level that remained significant even after additional adjustment for maternal dietary variables. Finally, higher adherence to the “semi-skimmed milk, yogurt” consumption pattern was associated with higher C15:0, DHA, total n-3 PUFA level and with lower LA, total n-6 PUFA levels and n-6/n-3 LC-PUFA ratio.

### 3.2. Associations between Maternal DP Patterns and Fatty Acid Levels in Cord RBC Membrane

Associations between maternal DP patterns and fatty acids level in cord RBC membrane are presented in Figure 3. Higher adherence to “cheese” consumption pattern was associated with greater C15:0 level and lower total n-6 PUFA levels throughout successive adjustment. Higher adherence to “reduced-fat DP” consumption pattern was associated with lower C17:0, ALA, total n-6 PUFA levels and n-6/n-3 LC-PUFA ratio and with higher DHA and total n-3 PUFA levels even after accounting for either maternal healthy dietary pattern or fish consumption during pregnancy. Lastly, higher adherence to the “semi-skimmed milk, yogurt” consumption pattern was associated with higher DHA, total n-3 PUFA levels but these associations did not remain significant after adjustment for maternal fish consumption during pregnancy. Greater adherence to the “semi-skimmed milk, yogurt” consumption pattern was associated with lower n-6/n-3 LC-PUFAs ratio. In overall, no associations were observed with OCFAs, LA and AA level.

### 3.3. Associations between Maternal DP Consumption and Fatty Acids Level in Colostrum

Associations between maternal DP patterns and fatty acids composition in colostrum are presented in Figure 4. Greater adherence to the “cheese” pattern was associated with higher C15:0 level and lower LA, total n-6 PUFA across all models.

Higher adherence to “reduced-fat DP” pattern was only associated with lower C15:0 level. Finally, greater adherence to “semi-skimmed milk, yogurt” pattern was associated with higher C15:0, C17:0 levels and with lower total n-6 PUFA and n-6/n-3 LC-PUFA ratio; however, associations with the total n-6 PUFA level and the ratio were not significant after accounting for either maternal healthy dietary pattern or fish consumption during pregnancy. No association was observed with AA and n-3 PUFAs (ALA, DHA).

In summary, fatty acids levels were associated with “higher-fat content” and “lower-fat content” DP consumption tertiles in the same direction as with “cheese” and “semi-skimmed milk, yogurt” consumption patterns, respectively (Appendix A). From our further analysis, maternal DP consumption was poorly associated with EPA and n-3 docosapentaenoic acid (DPA n-3) levels across perinatal biofluids (data not shown).

In our sensitivity analysis and for all studied associations, removing mothers with an energy intake lower than 1000 kcal/day or higher than 5000 kcal/day, those with gestational diabetes or who delivered prematurely did not affect our results (data not shown). In summary, no interaction of maternal overweight status on any association between maternal DP consumption and fatty acids levels in perinatal biofluids was observed.

## 4. Discussion

In the present study, greater adherence to a “cheese” consumption pattern during pregnancy was consistently associated with higher C15:0 level in maternal and cord RBC membrane, and in colostrum, and with lower total n-6 PUFA level in cord RBC membrane and in colostrum and lower LA in colostrum only. Results were less consistent for C17:0 across DP patterns and biofluids while both C17:0 and C15:0 have been suggested as biomarkers of dairy fat in the literature. Mainly, greater adherence to “semi-skimmed milk, yogurt” and “reduced-fat DP” patterns was associated with higher DHA and n-3 PUFA levels and lower n-6/n-3 LC-PUFA ratio in maternal and cord RBC membrane, and with lower total n-6 PUFA level in maternal RBC membrane.

From our findings, greater dairy fat consumption was associated with C15:0 in all perinatal biofluids analyzed herein even after accounting for other potential sources of C15:0. Our results support the plausible transfer of C15:0 from maternal circulation to the fetus via the placenta and to breastfed infants through breast milk [27]. This suggests that C15:0 might be a good biomarker of maternal dairy fat consumption during pregnancy in maternal, cord RBC membrane and colostrum. Dairy fat is a well-established source of C15:0 (and C17:0), hence C15:0 has been extensively used as a biomarker of dairy fat consumption in adult human biological samples but not yet in perinatal samples [18,19,20]. Beyond being a biomarker of dairy fat, C15:0, as an OCFAs, could be beneficial for many health conditions. This hypothesis is supported by studies showing that higher C15:0 and C17:0 levels in adult plasma phospholipids or RBC membrane was associated with lower occurrence of metabolic diseases such as type 2 and gestational diabetes, and cardiovascular disease [38,39,40,41,42]. From animal research, it was also suggested that C15:0 and C17:0 could play a role in the offspring brain development in utero and during breastfeeding period and hence on later cognitive development [26,27]. Studies on neurodegenerative diseases and mental disorder have brought supportive evidence on the role of C15:0 in neuronal membrane fluidity [43,44,45]. Yet, no observational study has been conducted during the early human developmental period to confirm these findings. From our study, associations with dairy fat consumption and C17:0 level are equivocal. Even if results are mostly in line with C15:0 results, the associations with C17:0 and DP consumption were not consistent across all DP patterns and perinatal biofluids. This may support previous evidence showing that C17:0 may not be a genuine biomarker of dairy fat consumption [22,46,47,48]. For instance, the C17:0 level was reported to be similar when comparing omnivorous, vegetarian, vegan and semi-omnivorous individuals biological fatty acid status [25]. Some potential endogenous synthesis pathways were proposed for both C15:0 and C17:0 [49]. Nevertheless, an animal study has shown that C15:0 could be less readily made endogenously than C17:0 [50]. Future studies investigating the role of dairy fat consumption in early developmental research are needed to confirm the relevancy of C17:0 over C15:0 as a biomarker of dairy fat consumption in perinatal biofluids.

From our results, higher adherence to a “cheese” consumption pattern was associated with lower LA and total n-6 PUFA levels in colostrum. This result is consistent with previous literature [15,17]. In a randomized trial, breastfeeding mothers assigned to a higher-fat content DP diet had a lower amount of LA in their breast milk compared with the control group and their depleting period [15]. Additionally, from an observational study, farming breastfeeding mothers with high intake of dairy full-fat milk during lactation, had lower LA and n-6 PUFA level in their breast milk [17]. Regarding our results on cord RBC membrane fatty acid level, higher adherence to a “cheese” and a “reduced-fat DP” was associated with lower total n-6 PUFA level and n-6/n-3 LC-PUFA ratio. To our knowledge, this is the first evidence on the association of maternal DP consumption with cord RBC, making it hard to draw any firm conclusion. Lastly, maternal greater consumption of lower-fat content DP, i.e., “semi-skimmed milk, yogurt” and “reduced-fat DP”, was associated with lower total n-6 PUFA level and n-6/n-3 LC-PUFA ratio in maternal RBC membrane. As a high n-6/n-3 LC-PUFA ratio has been linked to multiple chronic disease [51], further investigations are needed to evaluate whether the relatively lower n-6/n-3 LC-PUFA ratio in maternal RBC membrane in our study could be meaningfully associated with better maternal and offspring health outcomes. However, the relevance and meaning of this particular ratio has been seriously questioned, and a focus on the marine n-3 LC-PUFAs alone may be more important to health [52].

From our findings, higher adherence to the “semi-skimmed milk, yogurt” or “reduced-fat DP” patterns (but not to a “cheese” pattern) was associated with higher DHA and n-3 PUFA levels either in maternal or cord RBC membrane but not in colostrum. To explain these associations, as we are dealing with proportion of fatty acids rather than absolute values, a lower contribution of dairy fat to total fat intake is balanced by the contribution of other sources of fat, eventually richer in n-3 PUFAs such as vegetable oils or nuts and seeds, and those supplying substantial amount of DHA. Inconsistent associations with DHA level between colostrum and other perinatal biofluids may arise from additional origins of fatty acids in colostrum beyond maternal diet, i.e., de novo fatty acid synthesis in the mammary gland and mobilization from maternal adipose tissue storage [53]. As aforementioned, there is a lack of existing evidence to better document our findings.

In the current study, limitations are present. Inherent to the use of a self-reported FFQ, misreporting of maternal DP consumption can occur. In our sensitivity analysis, we attempted to account for it by excluding mothers with extreme energy intake but it did not change our results. Moreover, since our study population has mainly European ancestors, we cannot extrapolate our results to any population as fatty acid metabolism might differ by ethnicity [54]. DP consumption, and in particular higher-fat content DP consumption, is important in our study population [12]. This may increase our likelihood to observe an association compared with the population that has a low-consumption of higher-fat content DP. As all our tested associations were hypothesis driven and as the studied fatty acids are not independent of each other (% of total fat level in a given biofluid).

Our study is the first observational study to support the hypothesis of a placental transfer of OCFAs and their excretion in human breast milk. Additionally, our findings suggest that C15:0 over C17:0 may be a more reliable biomarker for maternal dairy fat consumption during pregnancy in perinatal biofluids but more studies are warranted to confirm those data. Our study also provides pioneering results on the association between maternal DP consumption during pregnancy and PUFA levels in three different perinatal biofluids. Having the fatty acid levels of three different perinatal biofluids, helps provide a comprehensive picture of how maternal DP consumption could affect the offspring fatty acid supply at different developmental stages. Lastly, maternal DP consumption was well characterized in terms of diversity, as our FFQ covered a wide variety of DP which helps us identify concisely different pattern of DP consumption.

## 5. Conclusions

To conclude, our results are in line with animal studies suggesting a potential transfer of C15:0 from maternal circulation to both fetal circulation through the placenta and the breastfed infant circulation through breast milk. The present study suggests that C15:0 over C17:0 may be a more reliable biomarker for maternal dairy fat consumption in maternal, cord RBC membrane and colostrum independently of other C15:0 dietary sources. Finally, consistently with previous studies, we found that greater dairy fat intake could be associated with lower LA levels in breast milk. As lower LA level in breast milk was associated with better cognitive outcomes in the breastfed offspring [55,56], further studies are warranted to determine whether maternal high consumption of full-fat dairy products could be beneficial for breastfed children brain development and hence, cognitive development.

## Figures and Tables

**Figure 1 nutrients-14-01636-f001:**
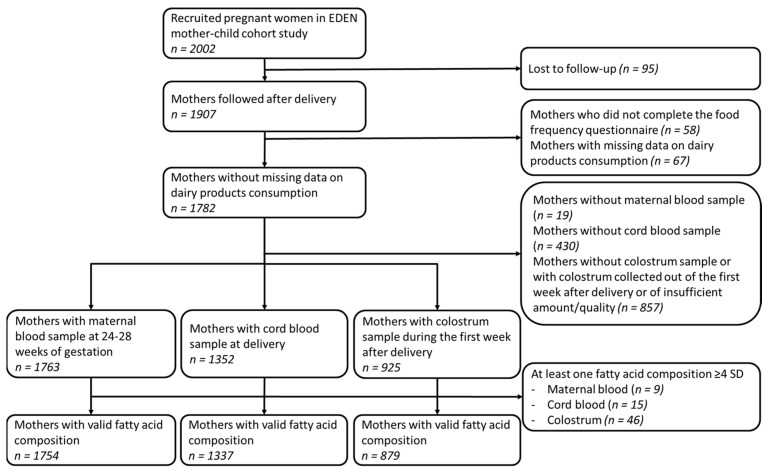
Sample selection flow chart.

**Figure 2 nutrients-14-01636-f002:**
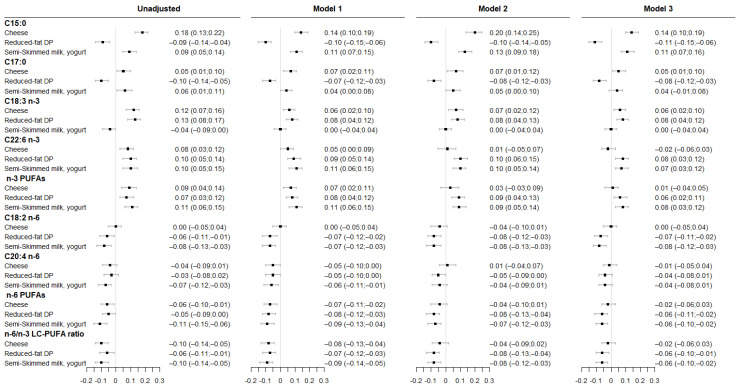
Associations between maternal DP patterns and fatty acid levels in maternal RBC membrane. Results are from linear regression models (β (95%CI)). Model 1 was adjusted for study center and maternal age at delivery. Model 2 was adjusted for model 1 covariates and for maternal “healthy” dietary pattern during pregnancy. Model 3 was adjusted for model 1 covariates and for the frequency of fish consumption during pregnancy. All DP patterns were studied simultaneously within each model.

**Figure 3 nutrients-14-01636-f003:**
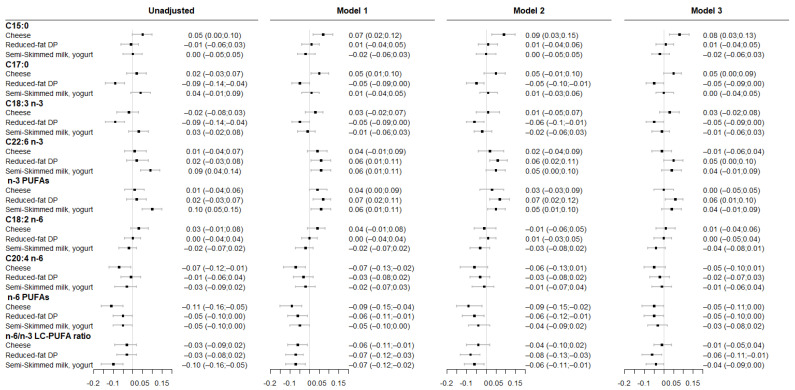
Associations between maternal DP patterns and fatty acid levels in cord RBC membrane. Results are from linear regression models (β (95%CI)). Model 1 was adjusted for study center and maternal age at delivery. Model 2 was adjusted for model 1 covariates and for maternal “healthy” dietary pattern during pregnancy. Model 3 was adjusted for model 1 covariates and for the frequency of fish consumption during pregnancy. All DP patterns were studied simultaneously within each model.

**Figure 4 nutrients-14-01636-f004:**
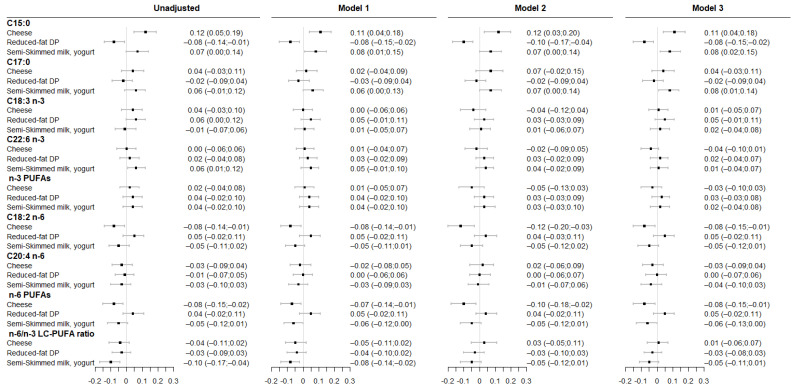
Associations between maternal DP patterns and fatty acid levels in colostrum. Results are from linear regression models (β (95%CI)). Model 1 was adjusted for study center, colostrum sampling day and maternal age at delivery. Model 2 was adjusted for model 1 covariates and for maternal “healthy” dietary pattern during pregnancy. Model 3 was adjusted for model 1 covariates and for the frequency of fish consumption during pregnancy. All DP patterns were studied simultaneously within each model.

**Table 1 nutrients-14-01636-t001:** Factor loadings on each maternal DP consumption patterns from principal component analysis (*n* = 1782).

Emmental, Gruyère, Comté, Beaufort in pieces	0.67	0.03	−0.11
Bonbel^®^, Babybel^®^, Gouda, Edam, Cantal, Tommes, Saint-nectaire, Reblochon	0.67	0.10	−0.20
Brie, camembert, pont-l’évêque, Munster, Vacherin, Saint-marcellin, Caprice des Dieux^®^	0.58	−0.02	−0.26
Emmental, Gruyère, Comté, Beaufort grated	0.49	−0.15	0.08
Goat cheese	0.49	−0.07	−0.27
Roquefort, blue cheese	0.47	−0.12	−0.19
Cottage cheese such as Tartare^®^ or Kiri^®^	0.42	0.20	0.14
Strained yogurt 0% fat	0.18	0.60	0.17
Reduced-fat butter	0.07	0.59	0.05
Reduced-fat cream	0.15	0.40	0.14
Skimmed milk	−0.02	0.38	−0.29
Full-fat milk	0.12	−0.16	−0.15
Cream	0.23	−0.52	0.11
Butter (added to a dish)	0.24	−0.54	0.26
Strained yogurt 20%, 40% fat	0.32	0.16	0.57
Yogurts (plain, flavoured, with fruits)	0.34	0.06	0.48
Semi-skimmed milk	0.01	−0.14	0.47
% explained variance	15	10	8
Component label	“Cheese”	“Reduced-fat DP”	“Semi-skimmed milk, yogurt”

**Table 2 nutrients-14-01636-t002:** Description of maternal characteristics in the study populations ^1^.

		Study Populations		
	EDEN Full Cohort (*n* = 2002)	Maternal Blood (*n* = 1754)	Cord Blood (*n* = 1337)	Colostrum (*n* = 879)
Age at delivery (years)	29 (5)	29 (5)	29 (5)	29 (5)
Educational attainment,%				
<high school diploma	29 (549)	27 (478)	27 (357)	22 (189)
high school diploma	18 (340)	18 (314)	17 (232)	15 (135)
2-year university degree	22 (414)	22 (384)	23 (305)	25 (214)
>2-year university degree	32 (607)	32 (565)	33 (432)	38 (334)
Monthly household income, %				
<1500 €	17 (327)	16 (287)	17 (225)	13 (111)
1500–2300 €	30 (568)	29 (506)	29 (390)	27 (240)
2301–3000 €	26 (501)	27 (466)	27 (354)	27 (237)
>3000 €	27 (517)	28 (485)	27 (358)	33 (285)
BMI before pregnancy (kg/m^2^), %				
<18.5	9 (161)	8 (146)	9 (120)	10 (83)
18.5–24.9	65 (1227)	66 (1134)	66 (866)	70 (602)
25.0–29.9	18 (330)	17 (298)	17 (221)	14 (120)
≥30.0	9 (166)	9 (149)	8 (103)	7 (61)
Gestational diabetes, %	6 (123)	6 (110)	6 (85)	6 (49)
Hypertensive disorders during pregnancy, %				
Gestational hypertension	3 (56)	3 (51)	3 (36)	3 (30)
Preeclampsia	2 (40)	2 (37)	2 (26)	1 (12)
Smoking during pregnancy, %	26 (484)	26 (450)	27 (351)	23 (199)
Any breastfeeding duration (months), %				
Never	27 (514)	27 (477)	29 (384)	0 (1)
<3	27 (502)	27 (464)	26 (346)	34 (296)
3–5	23 (435)	23 (398)	23 (301)	32 (284)
≥6	23 (440)	23 (410)	23 (303)	34 (294)

^1^ Values are the mean ± SD or % (*n*).

## Data Availability

The data underlying the findings cannot be made freely available for ethical and legal restrictions imposed, because this study includes a substantial number of variables that, together, could be used to re-identify the participants based on a few key characteristics and then be used to have access to other personal data. Therefore, the French ethics authority strictly forbids making these data freely available. However, they can be obtained upon request from the EDEN principal investigator. Readers may contact barbara.heude@inserm.fr to request the data. The analytic code will be made available upon request pending application and approval.

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
