# Peer review of "Associations of Maternal Consumption of Dairy Products during Pregnancy with Perinatal Fatty Acids Profile in the EDEN Cohort Study"

_nutrients, 2022, doi:10.3390/nu14081636_

Round 1

Reviewer 1 Report

This study compared the levels of OCFA in maternal and cord RBCs and in colostrum from a large cohort of women in France, and correlated these not only among themselves, but with dairy product use during pregnancy. They concluded that C15:0 appeared to be a better marker of dairy fat intake than C17:0 and that C15:0 is transferred from mother to baby both in utero and after birth through milk.

Overall, despite the tsunami of data presented, this report is helpful in answering the simple question of whether dairy FAs go from mother to baby or not. The paper could certainly be simplified greatly if the authors had just focused on the OCFAs in this report. The inclusion of the LC PUFAs in the same paper is, if nothing else, highly distracting and could be included in a second report.

Minor comments

Figure 1… all makes sense except “at least 1 fatty acid composition >4SD”. Do you mean at least one of the OCFA of interest with levels >4SD from the mean? Or ANY fatty acid in the sample?

Table 1. Perhaps put your Component Label above the column of numbers instead of at the end. Also, % of explained variance is not clear. Is Cheese 0.15% or 15%?

RBC membrane analysis… did you actually isolate the membranes per se, or did you measure total RBC FA composition and assume that all the FAs were in the membranes?

P9. “In overall [SUMMARY], fatty acids levels were associated with “higher-fat content” and “lower-fat content” DP consumption tertiles in the same direction than [AS?] with “cheese” and…” Also on P10, “In overall” should be either “Overall” or “In summary…”

P11. “As a high n-6:n-3 LC-PUFA ratio has been linked to multiple chronic disease [48], further investigations are needed to evaluate whether the relatively lower n-6:n-3 LC-PUFA ratio in maternal RBC membrane in our study could be meaningfully associated with better maternal and offspring health outcomes.” This statement should be qualified to say something like, “however, the relevance and meaning of this particular ratio has been seriously questioned, and a focus on the marine n3 LCPUFAs alone may be more important to health.” A good citation might be: https://pubmed.ncbi.nlm.nih.gov/29599053/

“Lastly, as we performed multiple tests, we cannot rule out some chance findings among our results.” That’s a real understatement! There must be 100s of correlations in different models included here. You might want to acknowledge this with a little more precision and suggest which findings you feel are the very clearest and strongest and not likely to be “chance”.

Additionally, our findings suggest that C15:0 MAY BE A MORE RELIABLE BIOMARKER  … THAN C17:0…” And if you want to repeat this in the Conclusions you might clarify the statement there as well.

“As lower LA level in breast milk was associated with better cognitive outcomes in the breastfed offspring [51, 52],”… was this also true for ARA, and how were EPA and DHA related with cognitive outcomes. Might be good to comment on more that just LA, unless the authors are willing to attend to the suggestion to refocus this paper only on OCFAs. In that case, this PUFA-related question becomes irrelevant. 

Author Response

This study compared the levels of OCFA in maternal and cord RBCs and in colostrum from a large cohort of women in France, and correlated these not only among themselves, but with dairy product use during pregnancy. They concluded that C15:0 appeared to be a better marker of dairy fat intake than C17:0 and that C15:0 is transferred from mother to baby both in utero and after birth through milk.

Overall, despite the tsunami of data presented, this report is helpful in answering the simple question of whether dairy FAs go from mother to baby or not. The paper could certainly be simplified greatly if the authors had just focused on the OCFAs in this report. The inclusion of the LC PUFAs in the same paper is, if nothing else, highly distracting and could be included in a second report.

Response: We thank the reviewer for the time allocated to review our manuscript and for giving us the opportunity to answer to the raised issues. We acknowledge that our work hypotheses need to be better clarified. Actually, our study relies on two hypotheses. First, as the reviewer mentioned, we hypothesized that C15:0 and C17:0 in cord blood and colostrum are biomarkers of maternal dairy fat consumption hence that C15:0 and C17:0 are transferred to the fetus and the breastfed infants. Second, we hypothesized that maternal high-fat content dairy products consumption could influence LA and LC-PUFAs levels and hence the n-6/n-3 LC-PUFA ratio in all perinatal biofluids. Indeed, some evidence supports maternal high-fat content dairy product consumption was associated with lower LA levels in colostrum (precursor of n-6 LC-PUFAs). As there is a competition in the synthesis of LC-PUFAs between n-6 and n-3 PUFAs families, changes in LA levels in biofluids and tissues may also affect the synthesis of n-3 PUFAs levels. Furthermore, from an animal study, higher dairy fat consumption was associated with higher n-3 LC-PUFAs levels in several tissues. It was suggested that dairy fat consumption may increase the conversion of ALA to n-3 LC-PUFAs by preserving ALA from β-oxidation; ALA being less preferentially oxidized compared with short- and medium-chain fatty acids sourced from dairy fat (Drouin et al., 2018). We acknowledge therefore that there is a plethora of findings reported in our manuscript. In aim to simplify the paper, we decided to remove EPA and DPA from our main results (hence from our figures) but rather briefly mentioned them in the Results section. We brought changes in the Introduction and Results section as follows:

“Evidence from an animal study has shown that having a diet with dairy lipids lead to higher n-3 LC-PUFAs levels in several tissues. Hence, it was suggested that short- and medium-chain fatty acids from dairy lipids consumption may be preferentially β-oxidized compared with ALA, that could favor the conversion pathway of ALA to n-3 LC-PUFAs [14].”

“Here, we hypothesized that C15:0 and C17:0 in maternal or cord blood and colostrum are biomarkers of maternal dairy fat consumption during pregnancy hence that C15:0 and C17:0 could be transferred to the fetus and the breastfed infants. Second, we hypothesized that maternal high-fat content dairy products consumption could influence n-6 and n-3 precursors and LC-PUFAs levels (through the modulation of the precursors conversion rate to LC-PUFAs) and hence the n-6/n-3 LC-PUFA ratio in all perinatal biofluids.”

“From our further analysis, maternal DP consumption was poorly associated with EPA and DPA levels across perinatal biofluids (data not shown).”

Minor comments

Figure 1… all makes sense except “at least 1 fatty acid composition >4SD”. Do you mean at least one of the OCFA of interest with levels >4SD from the mean? Or ANY fatty acid in the sample?

Reponse: We agree with the reviewer that this sentence needs more precision. About the outlier, as they were defined purposely to control the validity for the entire fatty acid composition of the biofluid, outliers were defined as any fatty acid levels >4SD including even those that we did not study in our manuscript. We have added more details in the methods section as follow:  

“For the present analyses, study populations were restricted to women …whom biological samples had a valid fatty acid composition, i.e. all fatty acid level<4SD in the biofluids.”

Table 1. Perhaps put your Component Label above the column of numbers instead of at the end. Also, % of explained variance is not clear. Is Cheese 0.15% or 15%?

Response: we thank the reviewer for raising the data entry error and for the suggestion. Indeed, we meant 15% of explained variance and not 0.15%. All % of explained variance were corrected accordingly in the Table 1. The reason why we have put the labels at the bottom of Table 1 is because the component labels are given a posteriori based on the items’ loadings on the component. 

RBC membrane analysis… did you actually isolate the membranes per se, or did you measure total RBC FA composition and assume that all the FAs were in the membranes?

Response: In the present study, RBC membranes were indeed isolated per se. We have clarified the sentence and specified the reference that detailed the method used to isolate RBC membranes as follows:

“RBCs membrane were isolated by centrifugation [31] and stored at -80°C until analysis.”

P9. “In overall [SUMMARY], fatty acids levels were associated with “higher-fat content” and “lower-fat content” DP consumption tertiles in the same direction than [AS?] with “cheese” and…” Also on P10, “In overall” should be either “Overall” or “In summary…”

Response: We thank the reviewer for noticing these wording issue. We have replaced “than” by “as” P9 and “In overall” by “In summary”.

P11. “As a high n-6:n-3 LC-PUFA ratio has been linked to multiple chronic disease [48], further investigations are needed to evaluate whether the relatively lower n-6:n-3 LC-PUFA ratio in maternal RBC membrane in our study could be meaningfully associated with better maternal and offspring health outcomes.” This statement should be qualified to say something like, “however, the relevance and meaning of this particular ratio has been seriously questioned, and a focus on the marine n3 LCPUFAs alone may be more important to health.” A good citation might be: https://pubmed.ncbi.nlm.nih.gov/29599053/

Response: We thank the reviewer for his/her valuable inputs in the discussion of the ratio. We have added the suggested sentence and reference to qualify our aforementioned statement.

“Lastly, as we performed multiple tests, we cannot rule out some chance findings among our results.” That’s a real understatement! There must be 100s of correlations in different models included here. You might want to acknowledge this with a little more precision and suggest which findings you feel are the very clearest and strongest and not likely to be “chance”.

Additionally, our findings suggest that C15:0 MAY BE A MORE RELIABLE BIOMARKER  … THAN C17:0…” And if you want to repeat this in the Conclusions you might clarify the statement there as well.

Response: We thank the reviewer for this relevant input. We acknowledge that the mentioned sentence is vague. As our tested associations were hypothesis driven and as the studied fatty acids are non-independent of each other (% of total fat level in a given biofluid), we consider that correction for multiple testing is irrelevant (https://pubmed.ncbi.nlm.nih.gov/2081237/). We believe that all our findings are likely not chance findings but we agree that among our tested associations, some have been more documented in the literature than others. Hence, we have removed the previous sentence and detailed our decision to not correct for multiple testing in the Method section as follows:

“As all our tested associations were hypothesis driven and as the studied fatty acids are non-independent of each other (% of total fat level in a given biofluid), we did not perform any correction for multiple testing.”

“As lower LA level in breast milk was associated with better cognitive outcomes in the breastfed offspring [51, 52],”… was this also true for ARA, and how were EPA and DHA related with cognitive outcomes. Might be good to comment on more that just LA, unless the authors are willing to attend to the suggestion to refocus this paper only on OCFAs. In that case, this PUFA-related question becomes irrelevant. 

Response: In the conclusion section, as no association were found between maternal dairy product consumption and ARA or EPA or DHA levels in breastmilk but only with LA, we focused our conclusion on potential benefit on children cognitive outcomes of a lower LA levels in breastmilk only, as a perspective of our study. To simplify the paper, we have removed findings on EPA and DPA as aforementioned.

Reviewer 2 Report

The authors investigated levels of C15:0 and C17:0 in maternal, cord RBC membrane and colostrum as biomarkers of maternal diary intake. It is a well-designed, interesting topic, however, the following comments and questions were raised:

  • Page 1: to preferential LCPUFA transport via placenta please put references here (Nr 3: Haggarty, 2010 and Nr5: Gil-Sanchez, 2010)
  • Page 2, study population: for study design reference 25 doesn’t seem to be proper (Gozzo S et al 1981, rat study). Did you mean here ref 10, 29 or 51?
  • Figure 1: flow chart: recruited mothers: 2002, lost to follow-up: 97, but mothers followed after delivery is 1907 instead of 1905. Please correct (either lost to follow-up is 95 or number of followed mothers are 1905).
  • Based on Table S1 about 6% of included mothers had gestational diabetes. It is clear that GDM disturbs fatty acid metabolism in both mothers and their infants. Did you compare fatty acid levels in the two group (healthy vs. GDM mothers)? Did you see any significant differences in the values of investigated/published FA values (C15:0, C17:0. ALA, EPA, DPA, DHA, n-3 PUFA, LA, AA, n-5 PUFA, n-6/n-3 LCPUFA)?
  • Maternal as well as fetal FA metabolism can be affected by several maternal diseases, like preeclampsia/eclampsia, obesity, (GDM), hypertension. How many mothers in this cohort suffered from these diseases? Have any of these affected fatty acid supply (mainly published FAs)?
  • According to the heatmap (Figure S1) no correlations were found between on the one hand maternal (RBC, colostrum) and on the other hand cord RBC C15:0 as well as C17:0 values. By contrast maternal cheese consumption was associated with greater C15:0 in cord RBC and maternal reduced-fat DP with lower cord C17:0 values. How would you elucidate this?

Author Response

The authors investigated levels of C15:0 and C17:0 in maternal, cord RBC membrane and colostrum as biomarkers of maternal diary intake. It is a well-designed, interesting topic, however, the following comments and questions were raised:

Response: We thank the reviewer for the time allocated to review our manuscript and for giving us the opportunity to answer to the raised issues.

Page 1: to preferential LCPUFA transport via placenta please put references here (Nr 3: Haggarty, 2010 and Nr5: Gil-Sanchez, 2010)

Response: We thank the reviewer for the relevant references. We added the mentioned references in our manuscript.

Page 2, study population: for study design reference 25 doesn’t seem to be proper (Gozzo S et al 1981, rat study). Did you mean here ref 10, 29 or 51?

Response: We thank the reviewer for noticing this oversight in the referencing. The reference is missing here, it should be Heude et al, 2016 ( https://pubmed.ncbi.nlm.nih.gov/26283636/). We have corrected the reference accordingly.

Figure 1: flow chart: recruited mothers: 2002, lost to follow-up: 97, but mothers followed after delivery is 1907 instead of 1905. Please correct (either lost to follow-up is 95 or number of followed mothers are 1905).

Response: We thank the reviewer for raising this calculation mistake. The right number should be 95 lost to follow-up. We have corrected the flow chart accordingly.

Based on Table S1 about 6% of included mothers had gestational diabetes. It is clear that GDM disturbs fatty acid metabolism in both mothers and their infants. Did you compare fatty acid levels in the two group (healthy vs. GDM mothers)? Did you see any significant differences in the values of investigated/published FA values (C15:0, C17:0. ALA, EPA, DPA, DHA, n-3 PUFA, LA, AA, n-5 PUFA, n-6/n-3 LCPUFA)?

Response: As only 6% of mothers had gestational diabetes in our study, we have rather evaluated the influence of removing mothers with gestational diabetes in our results in a sensitivity analysis than performing a stratified analysis. Based on the sensitivity analysis (mentioned page 5 and 10), there was no difference in our estimate regardless of the fatty acid and the perinatal biofluids considered, after removing mothers with gestational diabetes.  

Maternal as well as fetal FA metabolism can be affected by several maternal diseases, like preeclampsia/eclampsia, obesity, (GDM), hypertension. How many mothers in this cohort suffered from these diseases? Have any of these affected fatty acid supply (mainly published FAs)?

Response: We thank the reviewer for this relevant question. Following the reviewer suggestion, we performed an additional sensitivity analysis by removing mothers with gestational hypertensive disorders (only 5% of EDEN mothers) and no difference was observed. We added the percentage of mothers with hypertensive disorders in our descriptive table, mentioned about the sensitivity analysis in the method section and reported the results accordingly as follows:  

In the Method section: “As only 6 and 5% of mothers in our study samples had gestational diabetes or hypertensive disorders, respectively, we removed these mothers in our sensitivity analyses.”

In the Results section: “In our sensitivity analysis and for all studied associations, removing mothers with an energy intake lower than 1000 kcal/day or higher than 5000 kcal/day, those with gestational diabetes or hypertensive disorders or who delivered prematurely did not affect our results (data not shown).”

According to the heatmap (Figure S1) no correlations were found between on the one hand maternal (RBC, colostrum) and on the other hand cord RBC C15:0 as well as C17:0 values. By contrast maternal cheese consumption was associated with greater C15:0 in cord RBC and maternal reduced-fat DP with lower cord C17:0 values. How would you elucidate this?

Response: There are correlations in C15:0 levels between biofluids but the magnitudes of these correlations are moderate to small. C15:0 level in maternal RBC membrane is correlated with its levels in cord RBC membrane and in colostrum (r=0.10, p=0.0007 and r=0.34, p<0.0001, respectively).

C17:0 level in maternal RBC membrane is weakly but positively correlated with its level in cord RBC (r=0.19, p<0.0001) but not with its level in colostrum (r=0.09, p=0.11).

Round 2

Reviewer 1 Report

The authors have responded well to my suggestions. I have no further concerns.